# Biobased Materials for the Development of Biodegradable Slow-Release Fertilizers

Przemysław Boberski [1,2,*], Kamila Torchała [1], Hanna Studnik [1], Jan Wójcik [1], Marek Główka [1,2] and Nikodem Kuźnik [2,*]

1 Łukasiewicz Research Network—Institute of Heavy Organic Synthesis "Blachowania", ul. Energetyków 9, 47-225 Kędzierzyn-Koźle, Poland
2 Faculty of Chemistry, Silesian University of Technology, ul. M. Strzody 9, 44-100 Gliwice, Poland
* Correspondence: przemyslaw.boberski@icso.lukasiewicz.gov.pl (P.B.); nikodem.kuznik@polsl.pl (N.K.); Tel.: +48-507-940-570 (P.B.)

**Abstract:** According to the regulations of the European Parliament on fertilizer products, by July 2024, all substances used for coating fertilizers should be biodegradable. A series of coated fertilizers was prepared, which differed in the amount of applied coating layer. The core of the composition was granular ammonium nitrate, which contains 27% nitrogen. The effects of the amount of oil layers were examined. The article shows the results of IR testing and Iodine Value of the materials used. The coated fertilizer was evaluated for the release of nutrients under water conditions according to the standard ISO 21263, and the water samples were taken after every day for a 7-day period. The nitrogen content was analysed by elemental analysis. Microscopic pictures of the fertilizer composition were taken before release. The obtained product had potential controlled-release properties and was environmentally friendly. The tested material shows high potential as a component of a two-layer coated fertilizer. This type of fertilizer could be particularly useful in agricultural and horticultural applications.

**Keywords:** controlled released fertilizer; biodegradable polymers; biodegradable coatings



## 1. Introduction

The continuously increasing population results in an increasing demand for the amount of food produced. Therefore, technological solutions are sought that would improve the production process. This also applies to agriculture, where the main factor that would improve the yield of the crops obtained is the effective use of nutrients supplied to plants by producers in the form of fertilizers [1,2]. Common granular fertilizers are enveloped to slow the release of nutrients from them. The release of nutrients occurs gradually throughout the vegetation cycle, resulting in uniform plant growth [3–5].

A definite advantage of coated fertilizers is the fact that they are used once, which limits the amount of work done in the field. By using coated fertilizers, there is no risk of overfertilizing plants [6]. It is possible to adjust the fertilizer release profile to meet the requirements of a specific type of crop, soil or climatic zone by coating the base fertilizer with one, two or more layers of described material [7].

Plastics, as a ubiquitous class of synthetic polymer materials, are used in virtually all commercial and industrial sectors and are used as a coating material for fertilizers, which leads to further contamination of environment with so-called "micro-plastics" [8]. Today, there is a growing interest in smart fertilizers, particularly slow-release fertilizers, where the coating materials are both biobased and biodegradable. This is due to the legislative changes that entered into force in 2019.

In the case of fertilizer management, it is expected that not only the packaging in which the fertilizer is stored will be environmentally friendly but also that the fertilizer itself will be a completely biodegradable element, leaving no trace after a certain time.

It has been noticed that, particularly in last two years, the number of studies regarding biodegradable polymers for fertilizer application has significantly increased. Based on article reviews, most of the studies are focused on bio-based materials, such as starch, lignin, cellulose, alginates and chitosan, are the subject of investigations [9,10]. Non-natural origin polymers, such as polyhydroxybutyrate, polylactic acid and acrylic acid copolymers, are also reported as biodegradable coating materials [11].

In light of the recent Regulation of the European Parliament and Council (EU) 2019/1009 of 5 June 2019 laying down provisions on the making available on the market of EU fertilizer products, it is required that, by July 2024, all components of fertilizer and its packaging should be made of biodegradable substances. To this date, the EU Parliament has not pointed the approved test method for degradability of coating materials but the Directive lays down the following criteria: "the polymer has at least 90% of the organic carbon converted into carbon dioxide in a maximum period of 48 months after the end of the claimed functionality period" and that "it ultimately decomposes only into carbon dioxide, biomass and water".

Hemp oil, as an example of a natural material with a slight chemical modification, can slow the release of nutrients while being a material susceptible to biodegradation according to the yet unclear criteria of its evaluation [12]. Using the possibility of curing drying oils, characterized by high values of iodine number, were prepared a series of fertilizer and are the subject of this study.

This paper describes studies on the dependence of the nitrogen release profile from the prepared fertilizer in which a different amount of coating material was applied. The main goal of this work is to show the possibility of replacing nonbiodegradable materials in fertilizer products.

## 2. Materials and Methods

### 2.1. Materials

Granular ammonium nitrate with trademark "Salmag" (Grupa Azoty S.A., Tarnów, Poland) containing 27% nitrogen was purchased from Agrosimex Sp. z.o.o. in the form of 2–5 mm granules. Cold-pressed hemp oil was provided by Oleofarm. Cobalt salt of 2-ethylhexyl acid (Merck, Kenilworth, NJ, USA) and 2-ethylhexyl manganese salt (Thermo Scientific, Waltham, MA, USA) were used as catalyst for oil curing. Oil, catalyst and solvents were used directly as purchased without prior purification.

### 2.2. Methods

#### 2.2.1. Iodine Value Determination

The iodine number is a value that indicates the degree of unsaturation of an oil. The higher its value, the more unsaturations are contained in the structure of the compounds, i.e., double C=C bonds.

The studies on the determination of the iodine number were conducted in accordance with the PN-EN ISO 3961: 2018-09 standard "Vegetable and animal oils and fats—Determination of the iodine number" by titration with sodium thiosulphate solution using starch as an indicator. The procedure includes the preparation of solutions as well as the samples prior to determination.

#### 2.2.2. Fourier Transform Infrared Spectroscopy

The hemp oil was the subject of spectroscopic studies. Hemp oil is a raw material that contains many unsaturated bonds. The disappearance of the characteristic bands indicates changes in its structure, as a result of which, it hardens.

The tests using the FT-IR method were conducted on the PN-ISO 6286 standard "Spectrometry molecular absorption. Terminology, general information, apparatus" and test procedure BA-AG/PB-05 "General procedure for testing samples using FT-IR spectrometry (THERMO, Waltham, MA, USA)".

FT-IR spectroscopy was conducted in the transmission mode in the range of 4000–550 cm$^{-1}$ resolution 4 cm$^{-1}$ and 16 scans against air spectrum using KBr pellets containing about 1% m/m of the test substance. ATR-FTIR spectra for liquid samples were taken in reflection mode in the range of 4000–550 cm$^{-1}$ resolution 4 cm$^{-1}$ and 64 scans against spectrum air.

Apparatus:

- FT-IR Nicolet 6700 spectrometer with Omnic and TQ Analyst software (version OMNIC™ Professional 7 with advanced ATR correction and TQ Analyst EZ addition) by THERMO.
- hydraulic press (SPECAC, Kent, UK).
- pasting machine (SPECAC, Kent, UK).
- ATR attachment (SPECAC, Kent, UK), crystals of zinc selenide (ZnSe) with 45° shear angle.

*2.3. Preparation of Coated Fertilizer*

First, the fertilizer was sieved, collecting the fraction in the range of 2–4 mm in diameter. Subsequently, modified hemp oil was prepared. To this end, 3% m/m manganese and cobalt salts are added to hemp oil. Then, the granulated fertilizer was mixed for 0.5 h with the prepared oil. The fertilizer was transferred to the furnace and baked for 1 h at 130 °C. After being baked and cooled, the final fertilizer was obtained. The whole operation was repeated many times to obtain the desired amount of coating (% of coating) applied to the grain (Figure 1).

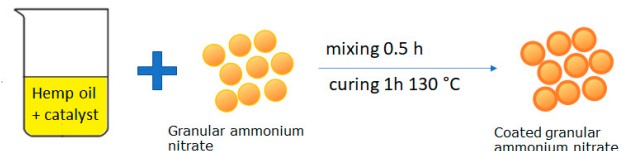

**Figure 1.** Illustrative scheme preparation of a fertilizer coated with polymerized hemp oil.

*2.4. Percentage Coating (Coating Efficiency)*

The percentage of fertilizer coating was determined by measuring the weight gain of the fertilizer in the coating process. For this purpose, the fertilizer sample was weighed before and after the encapsulation process after 24 h drying. The difference in mass compared to the initial mass is the fraction in which the fertilizer was coated. In this study, this difference was expressed as a percentage because the irregularity of the grains causes that the thickness of the shell may vary in some places.

$$\% \ coating = \frac{x_1 - x_0}{x_0} \times 100\% \qquad (1)$$

where $x_1$—mass of fertilizer after the coating process, $x_0$—mass initial fertilizer

*2.5. Determination Water Absorbency*

According to ISO 62 standards "Determination of the cold-water absorbency of plastics that do not contain water-soluble substances" consists in drying the material for 24 h, then cooling and weighing. The samples are placed in a vessel with distilled water for 24 h. After the specified time has elapsed, the samples are removed, dried and weighed again. The following formula is used for the calculation:

$$X = \frac{m_2 - m_1}{m_1} \times 100\% \qquad (2)$$

where $m_1$—initial mass of dry polymer, $m_2$—mass of polymer after soaking in water

*2.6. Microscopic Photos*

The surface morphology of coated fertilizer was examined using a digital microscope (DSX1000, Olympus, Tokyo, Japan) operating with DX10-XLOB10X lens at the magnifications of $200\times$ and $400\times$. Before analysis, the samples were cut to show the cross-section of the fertilizer. Pictures of three fertilizers were taken, and thicknesses of layers were measured using Olympus software (version DSX10-BSW-2).

*2.7. Biodegradation*

The manometric respirometry method, approved by the European Union guidelines [13] was used to determine the biodegradability of the samples of the modified hemp oil. This methodology was suitable for substances insoluble in the water keeping the dispersion of the test substance in the system at a constant level by continuous agitation.

The decomposition of the test substance was determined by the biological oxygen demand (BOD), i.e., the amount of oxygen required for the oxidation of organic compounds by microorganisms in the water environment. The tested modified oil, which was the only source of carbon for microorganisms, was applied directly to the bottles of the Lovibond® BOD sensor set with the inoculated medium. The initial concentration of the tested modified oil was $100 \, \text{mg/dm}^3$. The study was conducted in the dark for 28 days at $22 \pm 2 \, °C$.

The determination of the BOD was based on the measurement of pressure in the closed system. The microorganisms in the sample consumed oxygen and produced $CO_2$ that was absorbed by solid KOH. Negative pressure was created, which correlated directly with the BOD as a measurement value. Systems containing modified oil, systems containing reference material as well as systems that were blank tests containing only bacterial inoculum in the medium were prepared in at least two replicates. As a reference substance ethylene glycol was used for high biodegradability.

The percentage of biodegradation was calculated by measuring the amount of oxygen uptake in the bottle with the test substance corrected by the uptake in a parallel blank sample, expressed as a percentage of the theoretical oxygen demand (ThOD).

To determine the ThOD, the carbon and hydrogen content of the modified oil were determined by elemental microanalysis. The ThOD was then determined based on the formula according to the OECD 301 guidelines.

*2.8. Release Behavior of Coated Fertilizer in Water*

The method of determining the release of nutrients used is a modification of the method PN-EN 13266:2003 standard "Slow released fertilizers Determination of the release of nutrients—Method for coated fertilizers". According to the original procedure, the samples were taken after 1, 7, 14, 21 and 28 days. The amount of total nitrogen content was determined. Here, the test lasted a week and samples were taken every day.

## 3. Results

The main purpose of the research was to produce a slow-release fertilizer coated with a bio-based and biodegradable material. This work includes research results from preparation of coated fertilizers and characteristics of coating material. Fourier Transform Infrared Spectroscopy and determination of the Iodine Value were used as the main method to assess the material properties.

The effect of various mass share of polymer layer is discussed. The percentage share of the coating in relation to the fertilizer mass was determined based on the differences in the weight of the fertilizer after coating and before the processing.

The prepared fertilizer was characterized by PN-EN 13266:2003 standard to calculate the amount of nutrients released. The biodegradability of modified oil was determined according to the OECD 301 guidelines by the manometric respirometry method.

### 3.1. Iodine Value Determination

Hemp oil used in this study, as an example of drying oils, has a high iodine number value of 145 g $I_2$/100 g. This term describes vegetable oils with a high iodine value. Its high value in hemp oil is due to unsaturated fatty acids, and the most common are, linoleic, alpha-linolenic and oleic acids. This study has proved that such oils with high iodine number are able to polymerize via the autooxidative radical polymerisation route in relatively low temperature. Samples of fertilizer prepared by applying a thin layer of hemp oil with 3% catalyst at 130 °C, allowed to obtain a uniform, transparent and flexible shell.

This type of reaction is initiated and accelerated by UV-light and does not occur in complete darkness. The group of drying oils also includes such oils as: linseed, safflower, black cumin, sea buckthorn and others with a high iodine value. [14,15] It is worth noting that the addition of the catalyst slightly decreased the iodine value because it should be suspected that even in the cold, some reactions take place in the places of double bonds. (Table 1).

**Table 1.** Iodine values of the hemp oil samples.

| Sample | Iodine Value [g $I_2$/100 g] |
|---|---|
| Hemp oil | 145 |
| Hemp oil with catalysts | 136 |

### 3.2. Fourier Transform Infrared Spectroscopy

Raw hemp oil and polymerized hemp oil samples were examined using the FTIR technique. The analysis was focus to measure the change of absorbance of spectrum at wavelength directly related to unsaturation bands. Figure 2 shows a fragment of the IR spectrum covering the considered bands. As can be seen, the intensity band at 3007 cm$^{-1}$, responsible for the stretching vibrations of the olefinic CH double bond, decreases after the curing process.

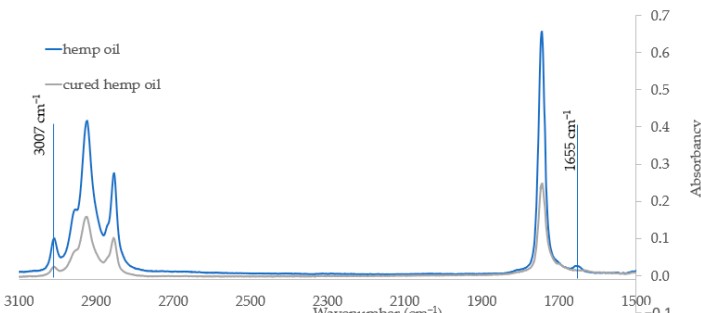

**Figure 2.** Part of the IR spectrum of hemp oil (blue) and polymerized hemp oil (gray) (range from 3100–1500 cm$^{-1}$).

The disappearance of the band occurring at 1655 cm$^{-1}$, coming from C=C unsaturated acyl groups, on the spectra of the cured oil indicates that the curing process takes place with the use of unsaturated groups. FT-IR spectroscopy can be successfully used in determination of whether the prepared oil is still "drying" or is already fully hardened. The disappearance of bands from groups related to double bonds indicates the completion of the process. The analytical results confirm obtaining of a hardened polymer material.

### 3.3. Water Absorbency

Water absorption is an important factor that allows to assess the material behaviour in contact with water. The amount of water absorbed by the coating material is determined according to the ISO 62 standards: "Determination of the cold-water absorbency of plastics that do not contain water-soluble substances".

In this test, a certain amount of material is placed in water at a constant temperature. After a specific time, the sample is pulled out of the water, dried and weighted. The water absorption is determined by the weight differences after and before the test. The measurements were done in three replications. The results allowed us to determine that the prepared material has an absorbency of an average of 2.3%.

This parameter is relevant because it allows to determine how much water can be absorbed by the polymer without disturbing its continuity, and in the case where more water is absorbed, the polymer layer loses its barrier properties, and the slow-release effect disappears. Water absorption at the level of 2.3% allows us to conclude that it has not a significant impact on the structure of the layer. By comparing with the value for other natural polymers, such as cellulose acetate—1.7% and nylon—6 1.3%. It could be said that the material absorbs a small amount of water [16].

### 3.4. Characterization Coated Fertilizer

The coating layer is obtained by means of auto-oxidative radical polymerization with the use of manganese and cobalt salts as a catalyst. The coating consists of chemical curing of the hydrocarbon chains, leading to physical hardening of the oil. This process takes place with the use of atmospheric oxygen, therefore multiple thin-layer coatings are preferred.

This process is conducted by mixing the uncoated fertilizer with hemp oil with the addition of catalysts (3% by weight). The fertilizer is then transferred to the oven where the hardening reaction takes place. The curing speed strongly depends on the process temperature, it cannot be too high to prevent oil degradation. On the other hand, too low a temperature leads to a long curing time, which is disadvantageous in terms of developing the technology for this process.

### 3.5. Percentage Coating

It is important to assess the quantity of polymers (unsaturated oils) used in the process. As the amount needed to operate efficiently and meet the standards for slow-release fertilizers increases, the last price of the product increases.

In the case of determining the amount of the applied coating, the increase in relationship between the fertilizer mass in the coating process was used. The fertilizer was weighed after each layer was applied. A series of fertilizers with a different proportion of the coating were prepared. The fertilizers where the oil layer constituted 2.5%, 5% and 10% were subject to the final evaluation. (Table 2).

**Table 2.** Percentage of coatings.

| Sample No. | Initial Fertilizer Mass [$m_1$] | Fertilizer Mass after Coating [$m_2$] | % Coating |
|:---:|:---:|:---:|:---:|
| 1 | 100 g | 102.51 g | 2.51 |
| 2 | 100 g | 105.12 g | 5.12 |
| 3 | 100 g | 110.07 g | 10.07 |

### 3.6. Release Behavior of Coated Fertilizer in Water

The prepared coated ammonium nitrate fertilizer with a different amount of cured hemp oil as a coating was evaluated for the release of nutrients in the water. The release test was conducted in accordance with PN-EN 13266:2003 standard "Slow released fertilizers Determination of the release of nutrients—Method for coated fertilizers". In this method, 5 g of coated fertilizer were placed in 250 g of water.

The sample is continuously agitated. Water samples were taken at certain times. A slight modification to the method was that samples were taken every day to determine the release rate in the initial phase of the test. For this purpose, the maximum amount of supernatant solution is decanted and then topped with the same amount of fresh water.

Figure 3 shows the relationship between the amount of nitrogen released and the time of the experiment. As can be seen, releasing rate strongly depend on amount of layer

used. It can be seen that fertilizers with 2.5% and 5% do not meet the requirements of the standard for not releasing more than 15% of the nutrients in the first 24 h. Only fertilizer with 10% of coating meets these criteria. In addition, this fertilizer did not release more than 15% of nutrients during the entire test period.

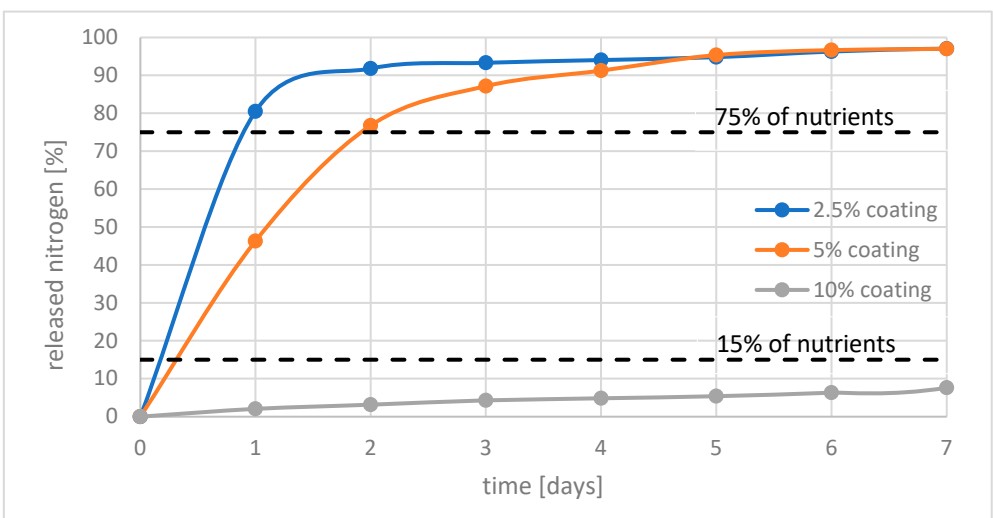

**Figure 3.** Release of nutrients within 7 days from polymer coated fertilizers with 2.5%; 5%; 10% of coatings.

Therefore, it can be concluded that using cured hemp oil as a coating is efficient when applied in minimum 10% by mass of fertilizer. In the case of fertilizers where a smaller amount of layer has been used, the effect of delayed release of nutrients is still observed. However, the release rate is too fast. It is promising to use cured hemp oil in a multilayered system.

### 3.7. Microscopic Photos

Microscopic photos (Figure 4), show the coated granular fertilizer. Photomicrographs show differences between prepared fertilizers. The photo shows the fertilizers with coating in the following amount: from the left 2.5%, 5% and 10%. The colour changes as the proportion of the coatings increases from light orange to brown to deep brown at 10% coatings. Looking at the photos from above, the coating seems to be even, there are no visible cracks or holes. Cross-sectional photos of the fertilizer show the thickness of the layers obtained.

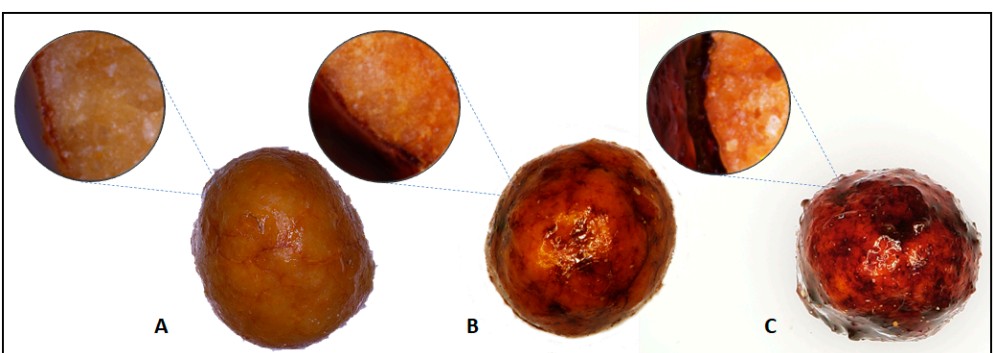

**Figure 4.** Images of coated fertilizer top and cross-section view (**A**) 2.5% coatings; (**B**) 5% coatings; and (**C**) 10% coatings.

With the increase of the coating mass, the layer thickness increases, which is confirmed by research based on the mass increase in the coating process. The measured average thickness was: 28.8 μm for 2.5% coated fertilizer, 38.2 μm for 5% coated fertilizer and 68.3 μm for 10% coated fertilizer.

### 3.8. Biodegradation

Assessment of the environmental impact of modified polymer shells, particularly in terms of biodegradability and toxicity, is nowadays an additional but crucial criterion apart from performance parameters. In certain industries, such as the food, pharmaceutical and cosmetics industries, it is necessary to use various composites and films, of a nontoxic and physiologically inert nature.

The study of the course of the biodegradation process may include assessing the degree of primary (initial) or total biodegradation. The first step is related to the appearance of intermediates called metabolites. Initial biodegradation occurs relatively quickly. The statutory value of the minimum susceptibility to degradation and the ability to remove is 80%. In subsequent reactions, further degradation of the metabolites may take place, leading to complete oxidation of the compound, and then complete biodegradation takes place.

According to the guidelines of the 301 OECD, it is believed that readily degradable organic compounds should under such conditions achieve a degradation of 60% as measured by the measurement of dissolved organic carbon (DOC) or oxygen consumption depending on the study methodology on 28 days. While organic substances are usually rapidly biodegradable during the three-week study, then in the case of more complex substances, such as polymers, even of natural origin, this time is insufficient to induce the production of appropriate enzymes in all microorganisms in the activated sludge, diverse in terms of origin. However, these studies can be used to predict the degradation time of such compounds in the environment.

The test results of biodegradation of the modified hemp oil over time, are presented in Table 3. Ethylene glycol was used as the reference substance. The activated sludge from the second outflow of the biological chamber of the municipal sewage treatment plant in Kędzierzyn-Koźle was used as the inoculum. Each time the reference material in the research achieved the intended degree of 70% biodegradation in less than 14 days.

**Table 3.** The results of the biodegradation test of the modified hemp oil sample.

| - | Biodegradation % | | | | |
|---|---|---|---|---|---|
| Time [days] | 0 | 7 | 14 | 21 | 28 |
| Modified hemp oil | - | 6.6 | 9.7 | 10.9 | 13.6 |

The results of oxygen biodegradation by means of manometric respirometry (OECD 301F) of modified oil showed the effect of chemical modification on the ability of their microbiological degradation. The use of hemp oil contributed to the achievement of the highest degree, above 13%, of biodegradation of this product. This value makes it possible to assume that used the material used will biodegrade within the time specified by the regulation. In European Union, Parliament has said that all used material has to biodegrade by 90%, within 48 months after releasing nutrients [9]. In this case, it is possible to assume that the whole material will biodegrade within about 8 months.

## 4. Discussion

The invention of efficient and economically viable coated fertilizers using biodegradable materials is something that fertilizer manufacturers are striving for, therefore it is difficult to find and cite all the solutions that are currently under development in this area. However, the materials that appear most frequently in the literature are cellulose [17], lignin [18] and chitosan [19]. The first two are natural polymers derived from wood processing and cannot be used without prior modification, e.g., esterification [20] because

of their poor mechanical properties. Furthermore, the medical use of chitosan due to its biocompatibility and antibacterial properties [21] makes its price too high to be used in fertilizers that should be cheap and available.

The group of oils with a high iodine number, including hemp oil, the so-called drying oils, seem to be promising natural resources to produce biodegradable coatings for controlled released fertilizers. Compared with recent research results, the use of hemp oil as a coating material is profitable because of its relatively low prices, abundance and the possibility of using without extra modifications.

The determination of the iodine number allows us to assess whether we are working with drying oil. When a catalyst has been added to the oil, a decrease in the iodine value can be observed. The FTIR spectra show a decrease in the intensity of the bands in the area of double bonds after the hardening process. Reactive areas of double bonds take part in the curing reaction, a decrease in the number of double bonds causes a decrease in the intensity of the characteristic bands.

Polymerized hemp oil as a coating material allowed for to formulate of a fertilizing composition with slow-release properties. The application of 2.5% of the coating extended the release of nutrients by 1 day, 5% of the coating extended this time to 4 days, while the using a 10% of the coating allowed for the release of only 8% on the seventh day of the test. 10% of the material used causes a significant reduction in the rate of nutrient release. This amount is acceptable to fertilizer producers.

Each increase in the quantity of the using material, will increase the final price of the product. The results for fertilizer with a 10% coating are promising and allow us to assume that this fertilizer will meet the requirements of the standard PN-EN 13266:2003 for coated fertilizers. In the microscope photos showing the cross section of the fertilizer, there is a visible coating, the thickness of which depends on the amount of material used. Thanks to the use of special image analysis software, this thickness can be measured. The unevenness of the coating thickness is a direct result of the irregularity of the fertilizer granules.

The use of biodegradable polymers is important in this case, as fertilizer is something that is directly introduced into the soil. Biodegradation of the material is crucial not to further pollute the farmland. The results of biodegradation tests at a level of 13.6% allow us to assume that polymerized hemp oil will degrade within 48 months as specified in the regulation of the European Parliament. However, it is important to make a review again. Fertilizers coated with biodegradable materials constitute an attractive and constantly evolving scientific discipline. There are undoubtedly many works to be done on the way to implement the technology; however, the obtained results provide clear insights into future solutions in the field of reducing environmental pollution with polymers

## 5. Conclusions

A new type of slow-release fertilizer was obtained by coating solid calcium ammonium nitrate with biobased raw material—hemp oil. According to the results of the OECD 301F biodegradability test, it can be presumed that the material will meet the criteria of incoming guidelines of testing fertilizers in terms of its biodegradability. However, the biodegradability assessment must be repeated according to the awaited test method, which will be approved by the European Parliament.

This study was focused on the issue of the dependence of the nutrient release profile with various amounts of coating material applied on fertilizer grain. The amount of coating material applied on solid fertilizer was measured by the weight gain of coated fertilizer and by measurement of the thickness of such a layer using an optic microscope with $200\times$ and $400\times$ magnification and integrated Image Analysis Software.

Hemp oil belongs to the group of drying oils, with Iodine Number of 145, due to which, it is possible to harden it and form a coating on the surface of the fertilizer granule. Due to the easy cultivation and rapid growth of hemp, its price is not excessive as in the case of safflower oil; however, it should be noted that there are also other cheaper oils, such as linseed oil, that could be used as a coating material.

It was shown that the use of cured plant oil—in this case hemp oil—allows a significant increase in the rate of nutrient release. Low water absorption values allow us to state that this material is not susceptible to water; it does not absorb too much water, and thus the structure does not break when it comes into contact with water.

Microscope images show that the "polymeric" hemp oil allows for the preparation of a smooth, flawless shell, and according to expectations, the highest release time and lowest leaching were observed for fertilizer with 10% coating material by weight. It can be assumed that this amount should be satisfactory to meet the standards for slow-released fertilizers.

**Author Contributions:** Conceptualization, P.B.; methodology, P.B., K.T. and N.K.; validation, P.B., K.T., J.W. and N.K.; formal analysis, P.B., K.T., H.S., J.W., M.G. and N.K.; investigation, P.B., K.T., H.S., J.W., M.G. and N.K.; resources, P.B.; data curation, P.B., K.T., H.S., J.W., M.G. and N.K.; writing—original draft preparation, P.B., K.T. and H.S.; writing—review and editing, P.B., K.T., J.W. and N.K.; visualization, P.B.; supervision, K.T. and N.K.; project administration, P.B. and K.T.; funding acquisition, P.B. All authors have read and agreed to the published version of the manuscript.

**Funding:** This research was co-financed by the Ministry of Science and Higher Education of Poland under Grant No. DWD/4/21/2020—06/003. The article processing charge was financed by the Ministry of Education and Science of Poland under Grant No. BB/22/10.

**Institutional Review Board Statement:** Not applicable.

**Informed Consent Statement:** Not applicable.

**Data Availability Statement:** Not applicable.

**Acknowledgments:** Thanks for support for Coordination Chemistry Group at Silesian University of Technology in Gliwice: https://twitter.com/CoorChem_PL (accessed on 7 June 2022).

**Conflicts of Interest:** The authors declare no conflict of interest.

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
