# Peer review of "Biobased Materials for the Development of Biodegradable Slow-Release Fertilizers"

_coatings, doi:10.3390/coatings12081215_

Round 1

Reviewer 1 Report

In this study, the authors report the studies on the dependence of the nitrogen release profile from the prepared fertilizer, in which a different amount of coating material is applied. And the coating material can significantly slowly released fertilizers. Some of the results are interesting, but there are still some questions. I think it can be published after making the following modifications.

1.      It seems that some items do not have units of measurement in the Table 2.

2.      Please check the case of English letters in reference

3.      Is 10% coating the optimal ratio? It is recommended to add other content data for comparison.

4.      Please check whether the "C-C bonds." in line 178 is correct.

5.      The serial numbers of the picture and annotation in Figure 4 should be consistent.

Reviewer 2 Report

This is new kind of research approach on 'biobased materials for the development of biodegradable SlowRelease Fertilizers'. However, there is always room for improvement and when something makes redundant it should be addressed.

1. there are some sentences that could be further imrpoievd by redrafting it. I have highlighted them in the text.

2. I am serious about methodology to be clear to reader. This portion is a bit unclear to understand. Calculation of fertilizer, coating material and its mass converting in percentage are good but not described why it was converted to percentages and what for that is useful in this study. Similarly, the estimation of iodine value and other parameters if so important in this coating fertilizer with biodegrade ables etc.should first be explained in introduction and based on their results it should be part of the results and discussion.

3. Results most part is expressing M&M section and it is relevant and shall be shifted. Here the results and its discussion is important to reader and shall be focused.

4. Conclusion shall be based on the results and the outcomes that why a particular oil type is to be the main focus of future fertilizer coating. It should be mentioned that there are some other plant oils of low price and could also be tried in future for commercial fertilizers.

Round 2

Reviewer 2 Report

The MS needs a through reading for minor English corrections.

The discussion needs to be extended with recent citation if available.
